# Continuous Diffusion Models Can Obey Formal Syntax

Jinwoo Kim [1]    Taylor Berg-Kirkpatrick [1]    Loris D'Antoni [1]

## Abstract

Diffusion language models offer a promising alternative to autoregressive models due to their global, non-causal generation process, but their continuous latent dynamics make discrete constraints—e.g., the output should be a JSON file that matches a given schema—difficult to impose. We introduce a training-free guidance method for steering continuous diffusion language models to satisfy formal syntactic constraints expressed using regular expressions. Our approach constructs an analytic score estimating the probability that a latent state decodes to a valid string accepted by a given regular expression, and uses its gradient to guide sampling, *without* training auxiliary classifiers. The denoising process targets the base model conditioned on syntactic validity. We implement our method in DIFFINITY on top of the PLAID diffusion model and evaluate it on 180 regular-expression constraints over JSON and natural-language benchmarks. DIFFINITY achieves 68-96% constraint satisfaction while incurring only a small perplexity cost relative to unconstrained sampling, outperforming autoregressive constrained decoding in both constraint satisfaction and output quality. DIFFINITY is open-sourced at `github.com/large-loris-models/Diffinity`.

## 1. Introduction

Diffusion-based language models are an increasingly attractive alternative to autoregressive generation, particularly in settings where bidirectional context or global revision is useful (e.g., infilling and long-horizon planning) (Gulrajani & Hashimoto, 2023; Kang et al., 2025; Lou et al., 2023; Jo & Hwang, 2025; Nie et al., 2025; Ye et al., 2025). Because a diffusion model denoises an entire sequence jointly across timesteps, it can in principle model non-causal dependencies and enforce global structure in ways that left-to-right autoregressive decoding cannot. This aspect makes diffusion a natural candidate for *structured generation*, where outputs must satisfy discrete syntactic requirements—such as JSON schema, mathematical formats, or regular expressions.

> We show that *continuous* diffusion models can be steered to satisfy formal syntax (regular expressions), using a training-free guidance mechanism.

**Why is this problem challenging?** In the autoregressive models, syntactic validity is typically enforced via *constrained decoding*, which filters next-token choices to ensure the final output is well formed (Geng et al., 2023). However, a continuous diffusion model never maintains an explicit discrete prefix or in general a plain text representation of the output sequence: each timestep operates on a continuous latent representation of the *entire* sequence. As a result, the core primitive behind constrained decoding—masking next-token choices that would make the current prefix uncompletable—does not transfer directly to continuous diffusion. Existing constrained-decoding adaptations for diffusion (Suresh et al., 2025; Mündler et al., 2025) therefore target specialized *discrete* diffusion variants that mimic autoregressive unmasking transitions (Nie et al., 2025; Ye et al., 2025) (though they do not require unmasking to happen left-to-right). Other approaches for steering diffusion models to satisfy constraints use classifier guidance (Dhariwal & Nichol, 2021; Song et al., 2020), which requires training an auxiliary model for each constraint. Because regular expressions induce nonlocal, discrete constraints, a classifier-free method for steering *continuous* diffusion models to obey formal syntax has remained out of reach.

**Our solution.** We address this gap by introducing a *training-free* guidance mechanism for continuous diffusion language models that targets *discrete* syntactic constraints. At a high level, our method mirrors classifier guidance (Dhariwal & Nichol, 2021): rather than modifying the base diffusion model, we add a guidance term during sampling that pushes trajectories toward regions of latent space that decode to syntactically valid text. The key difference is that we do not train an auxiliary classifier; instead, we ana-

[1]Department of Computer Science and Engineering, University of California-San Diego, San Diego, USA. Correspondence to: Jinwoo Kim <pl@ucsd.edu>.

*Proceedings of the 43rd International Conference on Machine Learning*, Seoul, South Korea. PMLR 306, 2026. Copyright 2026 by the author(s).

lytically compute the expected probability that a given latent state will decode to a sequence satisfying a target regular expression. Using the gradient of this probability to steer sampling yields two practical benefits: (1) **Training-free guidance:** The guidance score is computed analytically from the constraint, so applying a new regex does not require training (or storing) an additional classifier. (2) **Principled conditioning:** The resulting guidance approximates conditioning the diffusion sampler on syntactic validity rather than heuristically biasing token choices.

We instantiate our framework for regular expressions (regexes), which capture a wide range of practical syntactic constraints and are substantially more expressive than the constraints supported by prior non-classifier diffusion steering methods (Gulrajani & Hashimoto, 2023). Moreover, our guidance targets the *conditional distribution* of the base model given syntactic validity, enabling *constraint-aligned generation* in which samples remain faithful to the model rather than being distorted by ad hoc decoding heuristics. This stands in contrast to autoregressive constrained decoding, where enforcing validity can be at odds with preserving the original model distribution (Park et al., 2024; Parys et al., 2025; Anaya Gonzalez et al., 2025).

**Results.** We implement our method in DIFFINITY and apply it to PLAID, a a state-of-the-art *continuous* diffusion language model (Gulrajani & Hashimoto, 2023) in terms of performance.[1] Across structured-generation benchmarks (including regular versions of JSONSchemaBench (Geng et al., 2025) and natural-language regex constraints), DIFFINITY achieves high constraint satisfaction—up to 70% on complex JSON-schema regexes—while incurring only a small perplexity cost relative to unconstrained sampling. In addition, DIFFINITY is more reliable in practice than existing autoregressive constrained-decoding toolkits (Guidance Contributors, 2023; Willard & Louf, 2023), which can fail under finite token budgets (max_tokens) by stalling in low-probability parts of the automaton; we observe improvements in both validity and text quality against autoregressive models of comparable capacity.

**Contributions.** We make three contributions:

1. We propose a training-free guidance method that steers *continuous* diffusion language models to satisfy *discrete* regex constraints (§2).

2. We provide a theoretical justification showing that the resulting sampler targets the base model conditioned on syntactic validity (§3).

3. We implement DIFFINITY and demonstrate strong validity–quality trade-offs on structured-generation benchmarks, outperforming autoregressive constrained decoding in both reliability and text quality under finite token budgets (§4). DIFFINITY is open-sourced at github.com/large-loris-models/ Diffinity.

## 2. Steering Diffusion Models to Satisfy Regular Constraints

We now describe at a high level our approach to steering diffusion models towards generating outputs that satisfy the syntactic constraint imposed by a given regular expression.

### 2.1. A Crash Course on Diffusion Models for Text

Diffusion models—more precisely, denoising diffusion probabilistic models (DDPMs) (Ho et al., 2020)—generate data by reversing a gradual noise-addition process. We denote the data as $x$, where $x_T$ is pure Gaussian noise and $x_0$ is the "clean" data. Critically, in *continuous* text diffusion models like PLAID (Gulrajani & Hashimoto, 2023), $x_0$ is not discrete text but a continuous *embedding* of a text sequence—specifically, a matrix of embedding vectors, one for each token position. Because $x_0$ resides in a continuous space, we can model the generation process via Gaussian denoising. The "reverse" denoising step at each timestep $t$ is defined by sampling $x_{t-1}$ from a Gaussian distribution parameterized by the model:

$$x_{t-1} = \mu_\theta(x_t, t) + \sigma_t \epsilon \qquad (1)$$

In Equation (1), $x_t$ is the current noisy latent embedding, $x_{t-1}$ is the less-noisy latent to be computed, $\epsilon$ is noise sampled from the standard normal distribution $\mathcal{N}(0, I)$, and $\sigma_t$ is a fixed variance schedule. The term $\mu_\theta(x_t, t)$ represents the predicted mean of the posterior distribution given the current latent $x_t$. In practice, the model uses a neural network to predict the clean embedding $x_0$ from $x_t$, and $\mu_\theta$ is computed analytically as a weighted linear combination of $x_t$ and this predicted $x_0$.

Since the denoised output $x_0$ is still a continuous latent representation, obtaining discrete text requires a decoder Dec. This decoder maps the latent embedding $x_0$ to a sequence of probability distributions over the vocabulary—a matrix of unigram distributions, where the distribution at each position is independent of the others. The final output sentence $s$ is obtained by sampling from $\text{Dec}(x_0)$, where each token in $s$ is sampled independently from the corresponding unigram distribution at its position. Following previous work, we rely on this interpretation of the latent space as a decoder-induced unigram distribution to derive the expected probabilities discussed in the following sections.

---

[1]While there exist other diffusion models that display better performance (Nie et al., 2025; Ye et al., 2025), these are unmasking based discrete diffusion models and do not directly fit our framework.

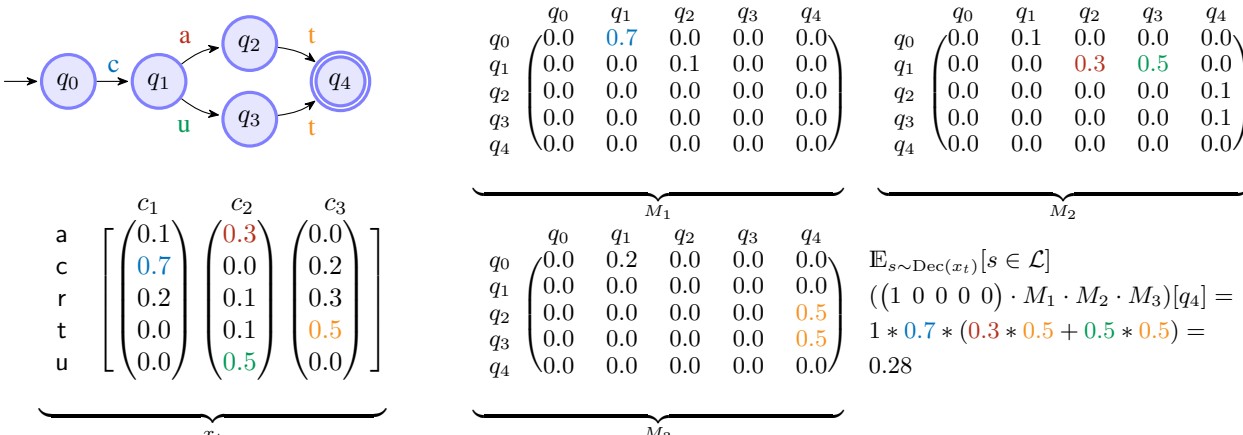

*Figure 1.* The automaton for the regex $\texttt{c(a|u)t}$ describing the regular constraint $\mathcal{L} = \{\texttt{cat}, \texttt{cut}\}$, the unigram distribution $\text{Dec}(x_t)$ defined by the current latent $x_t$ on a vocabulary $\{\texttt{a}, \texttt{c}, \texttt{r}, \texttt{t}, \texttt{u}\}$ of size 5, and transition matrices for this automaton and latent space for a sequence of length 3. Valid transitions inside the automaton and their probabilities are color-coded.

## 2.2. Denoising to Satisfy Regular Constraints

Our goal is to steer the denoising process defined by Equation 1 so that the generated output sentence $s$ belongs to a regular language $\mathcal{L}$.

Previous work has, to some extent, succeeded in guiding the diffusion process towards simple constraints such as "The sentence should start with the word 'Apple'," by simply maximizing the probability that the token "Apple" occurs at the start of the sequence (Gulrajani & Hashimoto, 2023). However, these existing approaches fail to extend to more complex structural constraints such as regexes. The standard approach to provide more complex structural guidance in other modalities is through *classifier guidance* (Dhariwal & Nichol, 2021). This approach steers the denoising process with input gradients from a pretrained control target classifier. However, training a classifier is often resource-heavy and difficult for syntactic constraints—e.g., a new classifier would need to be trained for each new regex.

The key idea in this paper is that, for syntactic constraints, we can construct an analogous steering mechanism via dynamic programming *without ever training a classifier*. More specifically, given the current latent $x_t$ and timestep $t$, the decoder Dec in most modern diffusion models (Nie et al., 2025; Jo & Hwang, 2025; Lou et al., 2023; Gulrajani & Hashimoto, 2023) are trained to emit the expectation of the *final* denoised output $\hat{x}_0$ as a unigram distribution $\text{Dec}(x_t)$ over the vocabulary. Then from the unigram distribution $\text{Dec}(x_t)$, we can analytically compute the probability that a sentence $s$ sampled from this distribution satisfies $\mathcal{L}$. This expectation becomes our "classifier" score, and we show how to compute its gradient.

Specifically, our modified steering procedure uses the fol-

lowing guided denoising step:

$$x_{t-1} = \mu_\theta(x_t, t) + \gamma \sigma_t^2 \nabla_{x_t} \log \mathbb{E}_{s \sim \text{Dec}(x_t)}[s \in \mathcal{L}] + \sigma_t \epsilon \tag{2}$$

In this equation, $\mu_\theta(x_t, t)$ represents the original model's predicted mean (the unconditioned update). The guidance term is defined by $\nabla_{x_t} \log \mathbb{E}_{s \sim \text{Dec}(x_t)}[s \in \mathcal{L}]$, which is the gradient of the log-expected probability that a sentence $s$ sampled from the unigram distribution $\text{Dec}(x_t) = \hat{x}_0$ satisfies the constraint $\mathcal{L}$, taken with respect to $x_t$. This gradient is scaled by $\sigma_t^2$ (the variance) to ensure the guidance signal is proportional to the noise level, and by $\gamma$, a guidance scale hyperparameter that controls the strength of the constraint enforcement.

Intuitively, Equation 2 shifts the center of the sampling distribution at each step toward regions of the continuous latent space that are expected to decode into valid sentences in $\mathcal{L}$. Maximizing this expected probability biases the reverse process to ultimately result in a valid discrete output $s$, while the $\sigma_t \epsilon$ term maintains the necessary stochasticity for high-quality generation. This formulation admits a natural interpretation through the lens of Langevin dynamics (Song et al., 2020), where the guidance term acts as an external potential. Specifically, the gradient of the expected probability exerts a force that steers the latent trajectory toward the manifold of embeddings that decode to strings in $\mathcal{L}$.

## 2.3. Computing the Expected Probability

To compute $\mathbb{E}_{s \sim \text{Dec}(x_t)}[s \in \mathcal{L}]$, we take advantage of the fact that any regular language $\mathcal{L}$ can be represented as a deterministic finite automaton (DFA). Our approach uses the DFA representing the regular constraint to build a series of *weighted transition matrices* $M_1, \cdots, M_n$ such that $M_k[i][j]$ represents the probability that the automaton will

transition from state $i$ to state $j$ by consuming the $k$-th token in the generated sequence, according to the unigram distribution given by the current latent.

*Example* 2.1 (Transition Matrices). Consider the regex 'c(a|u)t', which describe the regular language consisting of the words $\{cat, cut\}$, and a diffusion model with a vocabulary $\{a, c, r, t, u\}$. Figure 1 illustrates the DFA representation of this regular language (top-left), a distribution $\text{Dec}(x_t)$ that might occur from a latent $x_t$ during the diffusion process, and transition matrices $M_1$, $M_2$, and $M_3$ based on $\text{Dec}(x_t)$ when sequence length $l = 3$. Note how $\text{Dec}(x_t)$ is a unigram distribution: each column $c_i$ denotes probabilities that a certain token appears at the $i$-th position, independent of other positions.

Observe how in the first transition matrix we have that $M_1[q_0][q_1] = \text{Dec}(x_t)[c][1] = 0.7$, i.e., the probability that the automaton will take $q_0 \xrightarrow{c} q_1$ on the first token is the probability that c will appear at the first position according to the current latent point $x_t$. Similarly, for the second matrix: $M_2[q_1][q_2] = 0.3$ and $M_2[q_1][q_3] = 0.5$, as $\text{Dec}(x_t)[a][2] = 0.3$ and $\text{Dec}(x_t)[u][2] = 0.5$.

*Example* 2.2 (Computing Expected Probability). Given the transition matrices $M_1, M_2, M_3$, we start with $\mathbf{p}_0 = (1 \overbrace{0 \cdots 0}^{l-1})$—i.e., the automaton starts at the initial state with probability 1. The vector $\mathbf{p}_0 \cdot M_1 \cdot M_2 \cdot M_3$ represents the probability distribution over the states of the automaton after reading all three tokens according to the current latent $x_t$. We take the probability of $q_4$ (the only accepting state) in this distribution to obtain the expected probability. The bottom-right of Figure 1 illustrates the computation, where $1 * 0.7 * 0.3 * 0.5$ represents the probability that we take the path c, a, t, and $1 * 0.7 * 0.5 * 0.5$ represents the probability that we take the path c, u, t.

In particular, observe that $M_1[q_1][q_2] = 0.1$ because the token a can occur with probability $0.1$ in the first position according to $\text{Dec}(x_t)$. However, this probability gets multiplied by 0 when computing $(1 \quad 0 \quad 0 \quad 0 \quad 0) \cdot M_1$, as the probability of the automaton being in state $q_1$ is 0 at the first token—and thus does not contribute to the final expected probability (as it should, as a sentence starting with a violates the regex).

# 3. Formal Framework

First, we address the mismatch between regular-expression constraints and model tokenizers by constructing a vocabulary-aligned automaton (Section 3.1). Then, we present a dynamic programming algorithm that uses the vocabulary-aligned automaton to compute the expected probability that a sample satisfies the constraint (Section 3.2). Finally, we connect our approach to standard classifier guidance (Dhariwal & Nichol, 2021; Song et al.,

2020) and show how our approach approximates the desired conditional distribution (Section 3.3).

## 3.1. Tokenizer-Automaton Alignment

Before presenting the steering algorithm, we need to address a mismatch between the character-level definition of regular languages and the token-level representation used by diffusion language models. While a regular constraint specifies a set of valid strings, models generate sequences of tokens from a fixed vocabulary, and a single string may admit multiple valid tokenizations. To correctly compute expected probabilities and their gradients, the automaton representing the constraint must accept *all* token sequences whose concatenation forms a valid string; otherwise, valid probability mass would be systematically ignored.

Formally, if a regular language $\mathcal{L}$ accepts a string $x$, then the DFA $A$ for $\mathcal{L}$ must accept every tokenization of $x$. For example, for the string cat, if the automaton accepts the token sequence ca, t but not c, at, our approach in Section 2 would ignore the probability of the latter and thus compute an incorrect expected probability.

We introduce a function $\text{Align}(A, V)$ that constructs a vocabulary-aligned automaton $A_V$ from a given DFA $A$ and vocabulary $V$. The algorithm constructing $\text{Align}(A, V)$ (which is formalized in Section D and proven correct in Section E) first transforms $A$ into a character-level automaton $A_C$, whose transitions are labeled by individual characters. It then augments $A_C$ by adding transitions corresponding to multi-character tokens in $V$ whenever the underlying character path exists in $A_C$. For instance, if $V$ contains the token ca, the algorithm adds a transition $q_0 \xrightarrow{ca} q_2$ whenever a path labeled $c \rightarrow a$ exists from $q_0$ to $q_2$. Tokens that do not correspond to any valid character path (e.g., rt in Figure 1) are ignored. Finally, the algorithm removes character-level transitions not present in $V$, yielding a token-level automaton compatible with the model vocabulary.

## 3.2. Computing the Expected Probability

Having solved the alignment issue, Algorithm 1 formalizes the algorithm used to compute the expected probability $\mathbb{E}_{s \sim x_t}[s \in \mathcal{L}]$ given a latent $x_t$ and a regular constraint $\mathcal{L}$. Most concepts have already been illustrated in Section 2.2: $\mathbf{p}_0$ denotes the initial probability distribution over the states $Q_V$ of the (aligned) automaton, while lines 4 to 7 formalize the construction of the transition matrices $M_1, \cdots, M_l$.

Theorem 3.1 states the correctness of Algorithm 1.

**Theorem 3.1** (Expected Probability). *Let $A$ be a DFA that represents the regular language $\mathcal{L}$. Then Algorithm 1 returns the expected probability $\mathbb{E}_{s \sim x}[s \in \mathcal{L}]$.*

The proof is by induction on the fact that $\mathbf{p}_i$ denotes the

**Algorithm 1** Computing the Expected Probability

---

**Input:** DFA $A = (\Sigma, Q, q_0, \delta, F)$, latent $x$, seq. length $l$

1: $(Q_V, q_{V_0}, \delta_V, F_V) \leftarrow \mathsf{Align}(R, V)$

2: $\mathbf{p}_0 \leftarrow \overbrace{\begin{pmatrix} 1 & 0 & \cdots & 0 \end{pmatrix}}^{|Q_V|}$

3: **for** $k = 1$ to $l$ **do**

4: $\quad M_k \leftarrow 0_{|Q_V|, |Q_V|}$

5: $\quad$ **for** $(q, q') \in \delta_V$ **do**

6: $\quad\quad M_k[q][q'] \leftarrow \sum_{tok \text{ where } \delta(q, tok) = q'} \mathsf{Dec}(x)[tok][k]$

7: $\quad$ **end for**

8: $\quad \mathbf{p}_k \leftarrow \mathbf{p}_{k-1} M_k$

9: **end for**

10: **return** $\sum_{q \in F} \mathbf{p}_l[q]$

---

correct distribution over the states of the automaton at step $i$. The full proof is in the Appendix, Section E.

### 3.3. Connection to Classifier Guidance

Rather than deriving a new convergence proof, we observe that our approach is a training-free instantiation of *Classifier Guidance* (Dhariwal & Nichol, 2021; Song et al., 2020). In the standard framework, conditioned generation is achieved by modifying the score function (the drift term in the reverse SDE) with the gradient of a log-likelihood term:

$$\nabla_{x_t} \log \Pr(x_t \mid \mathcal{L}) = \nabla_{x_t} \log \Pr(x_t) + \nabla_{x_t} \log \Pr(\mathcal{L} \mid x_t) \tag{3}$$

In Equation (3), $\Pr(\mathcal{L} \mid x_t)$ is shorthand for $\int \Pr(\mathcal{L} \mid x_0) \cdot \Pr(x_0 \mid x_t) \, dx_t$, i.e., the probability that the final output $x_0$ predicted from the current latent $x_t$ satisfies the constraint $\mathcal{L}$. The first term $\nabla_{x_t} \log p(x_t)$ is provided by the base diffusion model, while the second term $\nabla_{x_t} \log p(\mathcal{L} \mid x_t)$—the "guidance"—is typically approximated by training a separate classifier network on noisy latents $x_t$.

Our method replaces this learned classifier with the analytical expectation derived in Algorithm 1:

$$\log \Pr(\mathcal{L} \mid x_t) \equiv \log \mathbb{E}_{s \sim \mathsf{Dec}(x_t)}[s \in \mathcal{L}] \tag{4}$$

Unlike a separate classifier which is trained to approximate $\Pr(\mathcal{L} \mid x_t)$, our analytical solution computes precisely $\int \Pr(\mathcal{L} \mid x_0) \cdot \Pr(x_0 \mid x_t) \, dx_t$. By using this analytical proxy, we leverage the robust theoretical foundations of guided diffusion without the cost or instability of training auxiliary classifiers. The convergence properties of our sampler thus follow directly from the standard results for score-based conditional generation established by Anderson (1982); Song et al. (2020).

## 4. Evaluation

We implement our approach as a tool DIFFINITY using the PLAID diffusion model (Gulrajani & Hashimoto, 2023) as

the base model. PLAID internally uses a 32-dimensional word2vec model as the latent space, and is equipped with a 1.3B transformer that interprets points in this space as unigram probability distributions (this transformer is trained alongside PLAID as part of its denoising network). DIFFINITY takes the gradient of this transformer composed with our implementation of the expected probability to obtain gradients on the word2vec latent space, then steers the diffusion process by adding the gradient term to the DDPM sampling equation that PLAID uses for denoising. At the end of the diffusion process, PLAID (and also DIFFINITY) samples using argmax from the final unigram distribution to produce the final concrete sentence. We reiterate that PLAID is a *continuous* diffusion model, and the decoding to a concrete sentence happens only once at the end of the denoising process; as opposed to discrete models (Lou et al., 2023; Nie et al., 2025) which decode to a discrete sentence at each step of denoising.

**Benchmarks** We consider 180 regular-expression constraints spanning two categories. **JSON**: 70 derived from JSONSchemaBench (Geng et al., 2025) (schemas converted to regexes when possible, then filtered for non-regular constraints and overly large tokenizer-aligned automata). **Natural Language**: 110 synthetically generated natural-language patterns covering Prefix, Suffix, Appearance, Between-$n$, Between (unbounded), and Word-length templates, each instantiated by sampling one or two words from the most common words in the English vocabulary (Tatman, 2018) and an integer parameter (e.g., position or distance). Details on the benchmark construction are in Section G.

We set the sequence length for all benchmarks to 64, which is a sufficiently long bound for generating sequences that satisfy our regex constraints. We run JSON generation benchmarks for 256 timesteps, because they result in complex automata that take a long time to compute the gradient over. For all other benchmarks, we run generation for 1024 timesteps. All other hyperparameters are set to the default values of the PLAID model (Gulrajani & Hashimoto, 2023).

**Metrics and Baselines** We report *constraint satisfaction rate* (the fraction of generated samples that satisfy the regex constraint) and *pass@10* (the fraction of benchmarks for which at least one of the first 10 samples satisfies the constraint), along with perplexity (geometric mean over all constraint-matching samples), and fluency (arithmetic mean over all constraint-matching samples). Perplexity is computed using LLama3.1-8B (Grattafiori et al., 2024) as the reference distribution, while fluency was measured on a scale of 0 to 100 by using Claude Sonnet 4.5-20250929 as an external LLM judge (prompts to derive fluency are adapted from Lee & Berg-Kirkpatrick (2025) and can be found in the Appendix, Section C).

*Table 1.* Constraint satisfaction and Pass@10 rates (in %) across JSON schema benchmarks. GPT2 models use Guidance (Guidance Contributors, 2023) for constrained decoding. Rates for DIFFINITY are reported with guidance scale $\gamma = 2.5$.

| Benchmark | GPT2-Medium | | GPT2-Large | | LLaDA+DINGO | | DIFFINITY | |
|---|---|---|---|---|---|---|---|---|
| | Sat. | Pass@10 | Sat. | Pass@10 | Sat. | Pass@10 | Sat. | Pass@10 |
| JSON | 75.3 | 97.1 | **77.0** | 98.6 | 63.3 | **100.0** | 68.4 | 91.4 |

*Table 2.* Combined comparison of constraint satisfaction (Sat., %), pass@10 (P@10, %), perplexity (PPL), and fluency (Flu.) across natural language benchmarks. GPT2-Large uses Guidance (Guidance Contributors, 2023) for constrained decoding. Perplexity and fluency are computed only on samples that satisfy the regex. Rates for DIFFINITY use guidance scale $\gamma = 2.5$.

| Benchmark | GPT2-Large | | | | LLaDA+DINGO | | | | DIFFINITY | | | |
|---|---|---|---|---|---|---|---|---|---|---|---|---|
| | Sat. | P@10 | PPL↓ | Flu.↑ | Sat. | P@10 | PPL↓ | Flu.↑ | Sat. | P@10 | PPL↓ | Flu.↑ |
| Prefix | 43.5 | 95.0 | 401.7 | 30.9 | **100.0** | **100.0** | 170.1 | 3.8 | 95.7 | **100.0** | 66.5 | **32.2** |
| Suffix | 17.5 | 55.0 | 70.3 | 31.9 | **100.0** | **100.0** | 219.2 | 3.4 | 96.8 | **100.0** | 59.3 | **36.8** |
| Appearance | 3.2 | 10.0 | 69.7 | **40.9** | **99.8** | **100.0** | 252.5 | 3.8 | 92.5 | **100.0** | 58.5 | 34.7 |
| Between-$n$ | 0.5 | 10.0 | 68.1 | **35.0** | **100.0** | **100.0** | 253.7 | 3.7 | 85.5 | 95.0 | **58.7** | 33.3 |
| Between (ubd.) | 3.5 | 10.0 | 71.2 | **47.1** | **95.5** | 95.0 | 234.5 | 3.6 | 93.8 | **100.0** | 57.4 | 33.3 |
| Word Length | 89.5 | **100.0** | 125.0 | **44.2** | **100.0** | **100.0** | 217.7 | 4.0 | 95.0 | **100.0** | 60.7 | 43.3 |

Our primary baselines are (*i*) constrained decoding for autoregressive models using the Guidance library (Guidance Contributors, 2023) with GPT2-Small, Medium, and Large, and (*ii*) constrained decoding for discrete diffusion using LLaDA-8B (Nie et al., 2025) with DINGO (Suresh et al., 2025), a constrained decoding algorithm for regular constraints on discrete diffusion. We also provide a short comparison against classifier guidance. The choice of GPT-2 models follows the original PLAID paper (Gulrajani & Hashimoto, 2023), which demonstrates that the capacity of the base PLAID model lies in between GPT2-Small and Medium. LLaDA is much stronger than PLAID (Nie et al., 2025) and is one of the strongest discrete diffusion models for text available; we use it as a baseline to illustrate the current capabilities and limitations of constrained decoding on discrete diffusion.

We evaluated DIFFINITY across different guidance scales $\gamma$ (Equation (2)). $\gamma = 2.5$ was most effective; reported statistics use $\gamma = 2.5$ unless otherwise noted. Classifier guidance was evaluated at $\gamma = 2.5$ as well. We also compare our computational overhead against PLAID's. The PLAID model also has native guidance capabilities that allow it to loosely model some of our regular expressions, so we make comparisons where applicable.

### 4.1. JSON Schema Benchmarks

Table 1 summarizes the results for the 70 JSON schema benchmarks (results for GPT2-Small are similar to those of other GPT2 models and presented in Section H). When generating text sequences of length $l$, unlike autoregressive models, the base PLAID model is trained to emit exactly $l$ tokens regardless of whether the sequence contains an <EOS> token or not. Thus to make a fair comparison, we

allow DIFFINITY to create arbitrary padding around the JSON schema and use the regex `.*schema.*` to constrain generation; LLaDA+DINGO are constrained using the same regex as well. For the GPT models, we use the non-padded regex (i.e., `schema` only). Alternative padding schemes are explored in the Appendix, Section H.

DIFFINITY has comparable performance to the constrained GPT-2 models and LLaDA+DINGO for JSON schema, being mildly outperformed by the GPT2-models, while achieving higher satisfaction rates but a lower pass@10 rate compared to LLaDA. Complex structured formats pose a fundamental challenge for smaller diffusion language models such as PLAID. Our JSON experiments demonstrate that DIFFINITY overcomes this limitation without the need of fine-tuning the model, enabling reliable generation of structured outputs outside the model's training distribution.

### 4.2. Natural Language Benchmarks

Table 2 summarizes the results for the natural language benchmarks. Due to space limitations, we present GPT2-Small and Medium's results in Section H. In terms of constraint satisfaction and pass@10 rates, DIFFINITY outperforms classifier guidance and autoregressive constrained decoding, achieving a very high average constraint satisfaction rate of 92.9% compared to 20.5% for GPT2-Large-GCD. LLaDA and DINGO achieve satisfaction rates of near 100%, but only do so with a significant cost in terms of perplexity and fluency.

The constraint satisfaction rates for the GPT2 models are low in these benchmarks primarily because they fall into the 'infinite repetition trap': for example, given a regex `[A-Za-z ] cat [A-Za-z ,.]*`, an autoregressive

model might continuously append characters to the first word if it finds that `cat` is an unlikely continuation to the first word it guessed until the `max_tokens` budget runs out. DIFFINITY is free of this problem, it will guide the denoising process towards an *entire, coherent sentence* that contains `cat` as the second word.

DIFFINITY also outperforms all GPT2 variants in terms of perplexity, consistently outperforming even GPT2-Large-GCD, with the smallest perplexity difference being 9.4 for the between (udb.) category. DIFFINITY's fluency (35.8) surpasses those of GPT2 Small (25.0) and Medium (32.0) and is comparable to that of GPT2-Large (37.2). We find this result remarkable; the original PLAID paper (Gulrajani & Hashimoto, 2023) indicates that the base PLAID model sits in between GPT2-Small and Medium in terms of perplexity.

Table 2 also illustrates that LLaDA+DINGO significantly *worsen* the quality of the base model's outputs: LLaDA was measured to have a geomean perplexity of 23.9 unconstrained, which increases sharply to 222.8 after constrained decoding with DINGO; the fluency rates also show that LLaDA+DINGO is generating incoherent samples.

DIFFINITY, on the other hand, does not negatively affect the performance of the base model: the average perplexity of all samples generated by DIFFINITY for the natural language benchmarks is 60.0 while the average fluency is 35.8 at scale 2.5. Our experiments indicate that the base PLAID model displays 61.6 perplexity and 46.0 fluency for unconditioned generation under the same hyperparameters (sequence length of 64 with 1024 diffusion timesteps). This shows that the **perplexity gap** induced by DIFFINITY is **virtually nonexistent** at scale 2.5, while the **fluency gap** is also **relatively low**—confirming that our approach in Section 2 allows diffusion models to obey formal syntax while preserving information from the original distribution. We once again note that this result is remarkable, showing that DIFFINITY outperforms existing discrete diffusion + constrained decoding approaches, *despite the fact* that discrete diffusion models show much better performance unconstrained. We provide an additional experiment that showing that DIFFINITY preserves the underlying probability distribution in the Appendix, Section I.

### 4.3. Comparison with Classifier Guidance

We briefly investigate how effective standard classifier guidance is on our regular expression benchmark. We train three versions of classifiers: (*i*) a direct-input classifier given full access to model internals, including the hidden state of the PLAID transformer (**Full**); (*ii*) a direct-input classifier with access only to the denoising latent of PLAID (i.e., $x_t$ in Equation (3) (**Latent only**), and (*iii*) a variant of the latent-only classifier where the denoising latent is passed through the backbone denoising transformer in PLAID (**With back-**

**bone**) (see Section F for details). We compare different configurations to make sure that classifier performance is not limited due to the type of information supplied. The table below reports satisfaction and pass@10 rates for the three different classifiers over our benchmarks.

|  | **Full** | | **Latent only** | | **With backbone** | |
|---|---|---|---|---|---|---|
|  | Sat. | P@10 | Sat. | P@10 | Sat. | P@10 |
| JSON | 0.00 | 0.00 | 0.00 | 0.00 | 0.00 | 0.00 |
| Natural | **2.27** | **10.00** | 1.45 | 8.18 | 0.14 | 0.91 |

Classifier guidance shows very low satisfaction rates for all benchmarks, reaching 0% for our JSON benchmarks, despite the fact that the classifier was trained relatively accurately, achieving an area-under-the-curve of 0.81, 0.74, and 0.79 respectively. In particular, we observed that classifier guidance was effective in steering the trajectory towards samples that score higher on the classifier: for example, average classifier score increased from 0.4 to 0.6 for the best-performing full classifier. However, the increase in classifier score *did not* translate to a significant increase in constraint satisfaction, which is a binary metric over a concrete structural constraint. On the other hand, DIFFINITY successfully optimizes for binary constraint satisfaction, by directly increasing the expected probability of constraint satisfaction as opposed to a proxy score learned by a classifier.

### 4.4. The Effect of Different Guidance Scales

We study how the guidance scale trades off constraint satisfaction and generation quality in DIFFINITY. Figure 2a shows how varying the guidance affects satisfaction rates, perplexity, and fluency.

As expected, increasing the guidance scale monotonically improves constraint satisfaction. The effect is particularly pronounced for JSON, where satisfaction rises from 30.4% at scale 1.0 to 68.4% at scale 2.5. We conjecture that the relatively large increase in satisfaction rate for JSON is because well-formed JSON is relatively further from the training distribution of the base PLAID model: at low guidance scales the diffusion trajectory frequently drifts into syntactically invalid regions, whereas stronger guidance anchors the reverse process to the JSON schema.

Surprisingly, stronger guidance does not degrade language quality. On the contrary, perplexity measured with Llama-3.1-8B decreases substantially, from 73.6 at scale 1.0 to 60.0 at scale 2.5. Fluency exhibits only a mild drop (from 39.4 to 35.8), indicating that the improvement in structural correctness is not achieved by sacrificing readability.

We hypothesize that this counterintuitive perplexity improvement arises because higher guidance suppresses unstable denoising trajectories that violate the regex early and subsequently require large updates. By constraining the diffusion

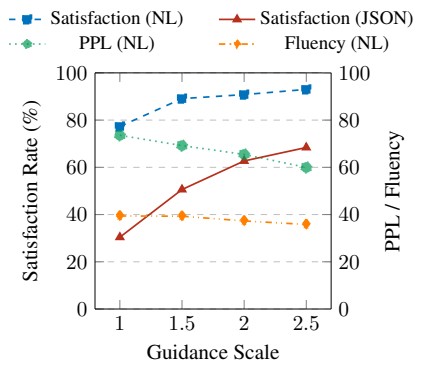

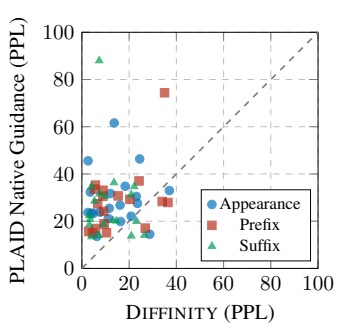

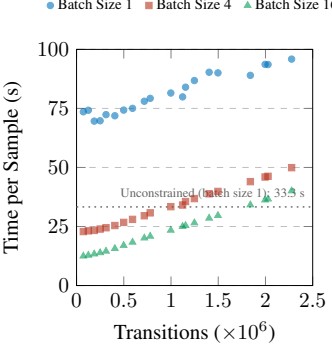

*(a)* Guidance-scale trade-offs for satisfaction, perplexity, and fluency.

*(b)* Minimum perplexity (PPL) comparison: DIFFINITY vs. native PLAID guidance.

*(c)* Generation latency vs. automaton size.

*Figure 2.* Guidance-scale trade-offs, PLAID perplexity comparison, and computational overhead.

path to remain close to the target regular language, DIFFINITY produces sequences that are both syntactically valid and more predictable under an external language model.

### 4.5. Comparison with PLAID

**Generation quality.** PLAID provides native guidance mechanisms that enable *token-level* control over simple syntactic properties, such as increasing the probability that a specific token appears at a given position, by injecting gradients of token log-probabilities during the denoising process. However, these mechanisms do not extend to structured constraints such as regular expressions. In particular, PLAID operates at the tokenizer level and is therefore incompatible with regex constraints due to the automaton–token alignment problem discussed in Section 3.1.

If we temporarily ignore this limitation, PLAID can express a strict subset of our natural-language benchmarks—namely the PREFIX, SUFFIX, and APPEARANCE categories. We therefore compare PLAID and DIFFINITY on this common subset, using a guidance scale of 2.5 for both methods. PLAID achieves an average satisfaction rate of 59.4% on these benchmarks, whereas DIFFINITY reaches 92.9%, demonstrating that automaton-level guidance is substantially more effective than token-level control even when both are applied at comparable strength.

Figure 2b reports a scatter plot of the minimum-perplexity samples obtained for benchmarks that both PLAID and DIFFINITY could solve. The scatter plot shows that DIFFINITY is advantageous in generating lower-perplexity samples, where DIFFINITY's minimum perplexity is on average 10.4 compared to 25.7 for PLAID. PLAID does have a slightly lower average perplexity for *all* generated samples that meet the constraint, at 55.65 compared to 60.27 for DIFFINITY; we attribute this difference to the fact that DIFFINITY has a much higher constraint satisfaction rate.

**Computational overhead.** We next examine the computational overhead introduced by DIFFINITY. Figure 2c plots per-sample generation latency for 20 representative benchmarks, across batch sizes of 1, 4, and 16, as a function of the number of transitions in the aligned automaton.

Overhead scales roughly linearly with the number of transitions, where a main element of the overhead is caused by Line 6 of Algorithm 1 which computes a dense sum over all automaton transitions. Although this computation is parallelized on the GPU, our automata contain a large amount of transitions (71735 for the smallest automaton, from the regex [A-Za-z]+ to [A-Za-z .,]* from the prefix category), and thus computing a dense sum over the transition matrix exerts significant pressure on memory bandwidth leading to overhead. The 71735-transition regex required 73.6 seconds to generate a sample at batch size=1, compared to 33.3 seconds for unconstrained generation on PLAID under the same constraints. Overall, DIFFINITY incurred a latency of around 2 to 3 times higher compared to unconditional generation, with higher latencies incurred by automata with more transitions. We do note that an intuitively 'complex' regex will not necessarily result in an automaton with a higher number of transitions: for example, wildcard matches create transitions equal to the number of tokens inside the tokenizer, and thus a simpler regex such as .* to .* .* and .* can result in a significantly higher-transition automaton compared to a more complex but constrained JSON schema regex.

In order to take advantage of GPU capabilities, our current implementation of DIFFINITY phrases most operations as matrix multiplications, causing memory overhead in DIFFINITY through two large matrices: *(i)* the matrix for computing the dense sum over the unigram distribution $\text{Dec}(x)$, which size scales linearly to the number of transitions inside the automaton, and *(ii)* the per-token-position transition matrices, each of which sizes scale with the square of the number of

states inside the automaton. For very large automata, such as those containing more than 3 million transitions, these matrices can consume significant amounts of memory, more than the base diffusion model itself.

While DIFFINITY does incur both latency and memory overhead, our primary goal in this work is to establish that diffusion models can reliably obey formal syntactic constraints—often more robustly than autoregressive approaches, as reflected in our perplexity and fluency results. Designing memory-efficient data structures and custom gradient kernels for Algorithm 1 is a promising direction for future work, and we expect optimizations like those used in constrained decoding (Park et al., 2025) to reduce the overhead of DIFFINITY by one to two orders of magnitude.

## 5. Related Work

Diffusion models, including DDPMs (Ho et al., 2020) and Variational Diffusion Models (Kingma et al., 2021), were originally developed for continuous data and have recently been adapted to text using both continuous and discrete paradigms. Continuous variants Li et al. (2022); Gulrajani & Hashimoto (2023); Jo & Hwang (2025) perform diffusion in a continuous latent space that is decoded to text; for example, PLAID diffuses in a 32-D word2vec space and decodes to a unigram distribution over sentences.

Discrete variants operate directly on token sequences by defining the forward noise-adding process as a probabilistic transition matrix (Lou et al., 2023). Masking diffusion models (D3PM (Austin et al., 2021), Dream (Ye et al., 2025), LLaDA (Nie et al., 2025)) introduce a special mask token and corrupt sequences by replacing tokens with masks, while others such as SEDD-Uniform (Lou et al., 2023) instead replace tokens with other vocabulary items.

Several approaches support constrained decoding for discrete masking diffusion models. DINGO (Suresh et al., 2025) enforces regular constraints, and Mündler et al. (2025) supports context-free constraints (which reduce to regular constraints for bounded length). These methods filter out unmaskings that must violate the constraint, but are limited to masking-based discrete models and distort the underlying model distribution. They do not apply to continuous diffusion models (our target) or mask-free discrete models such as SEDD (Lou et al., 2023).

Our work focuses on *continuous* diffusion models (PLAID in particular), modifying the denoising equation for continuous diffusion models à là classifier guidance. Because classifier guidance in theory is applicable to discrete models as well, we expect to be able to apply our approach to discrete diffusion models by developing a variant of Algorithm 1 that efficiently computes the expected probability for latent spaces of discrete models. We leave this idea as

an interesting venue for future work.

DIFFINITY currently only supports regular constraints, but in theory can be used to support context-free constraints as well, as bounding the length of a context-free language results in a regular language. Mündler et al. (2025) investigates how to impose these length constraints on a context-free language efficiently; extending these ideas to compute expected probabilities over length-bound context-free languages would be an interesting venue for future work.

The standard approach for steering diffusion models is *classifier guidance* (Dhariwal & Nichol, 2021; Song et al., 2020), which relies on a separately trained classifier. Our method follows the same principle but replaces the classifier with Algorithm 1, which computes the exact expected probability induced by a regular constraint.

PLAID (Gulrajani & Hashimoto, 2023) also supports limited steering using token-level probability gradients (e.g., enforcing the presence or absence of specific tokens at fixed positions). Such methods cannot express *global or sequential constraints* (e.g., 'Orange comes after Apple'), which require tracking dependencies across the full sequence.

Grammar-constrained decoding (GCD) (Geng et al., 2023) and its many implementations (Guidance Contributors, 2023; Park et al., 2025; Ugare et al., 2024; Willard & Louf, 2023) enforce syntactic constraints for autoregressive models by filtering out tokens that are guaranteed to eventually result in a string that violates the given grammar. GCD distorts the model distribution and can lead to the infinite repetition trap for complex grammars (Park et al., 2024). While recent sampling methods reduce this distortion (Park et al., 2024; Parys et al., 2025; Anaya Gonzalez et al., 2025), their benefits mainly arise when drawing hundreds of samples for the same task. As demonstrated in our evaluation, DIFFINITY targets the desired conditional distribution and does not suffer from the infinite repetition trap.

## 6. Conclusion

We presented DIFFINITY, a method for steering continuous diffusion language models to satisfy formal syntactic constraints specified by a finite automaton (or a regex). DIFFINITY guides diffusion sampling using gradients of the *exact* automaton acceptance probability under the model's current latent distribution, avoiding training a separate classifier. Using PLAID as the base model, we find that DIFFINITY is capable of successfully enforcing complex regular constraints while maintaining competitive generation quality.

While DIFFINITY's current computational overhead is relatively high compared to other methods, we believe existing work on succinctly representing automata (Park et al., 2025), can lift to DIFFINITY; a promising direction of future work.

## Acknowledgments

Supported, in part, by the National Science Foundations under grants CCF-2146151, CCF-2546822, CCF-2506134, CCF-2446711, and CCF-2422214; the Schmidt Foundation; by a Microsoft Faculty Fellowship; and a grant from the Korea Foundation for Advanced Studies. Any opinions, findings, and conclusions or recommendations expressed in this publication are those of the authors, and do not necessarily reflect the views of the sponsoring entities.

## Impact Statement

This paper presents work whose goal is to advance the field of Machine Learning. There are many potential societal consequences of our work, none which we feel must be specifically highlighted here.

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

## A. Hardware and Software

Our experiments were conducted on a Ubuntu 20.04.5 LTS server equipped with a AMD EPYC 7282 CPU (16-cores at 2.8GHz) and NVIDIA RTX A6000 GPUs. We ran all experiments using a single GPU. Our implementation is based on Python 3.10.19, PyTorch 2.0.1 with CUDA 11.8, and Flash Attention 1.0.9.

## B. Model Checkpoint

We use the following models in our evaluation:

- **PLAID**: From the PLAID github repository `https://github.com/igul222/plaid` (commit `e7d5cf8`)

- **GPT2**: Radford et al. (2019) with specific models being:

    - **GPT2-Small**: `https://huggingface.co/openai-community/gpt2` (commit `607a30d`)
    - **GPT2-Medium**: `https://huggingface.co/openai-community/gpt2-medium` (commit `6dcaa7a`)
    - **GPT2-Large**: `https://huggingface.co/openai-community/gpt2-large` (commit `32b71b1`)

- **Llama-8B**: Llama-3.1-8B (Grattafiori et al., 2024): `https://huggingface.co/meta-llama/Llama-3.1-8B` (commit `d04e592`)

- **Claude**: Claude Sonnet 4.5-20250929.

## C. Fluency Evaluation Prompt

We use the following prompt when querying Claude Sonnet 4.5 for fluency, for the generated sample `sample`.

```
"""Read the text below.  Then, indicate the fluency of the text, on a scale from
1 (poor fluency) to 100 (excellent fluency).  In your assessment, consider factors
such as grammatical correctness, naturalness of language, and overall smoothness.

Text to evaluate:  <Text> {sample} </Text>

On a scale from 1 (poor fluency) to 100 (excellent fluency), how fluent is this
text?  Respond in JSON format with two fields:  - "score":  an integer from 1 to
100 - "reasoning":  a brief explanation (1-2 sentences)

Example response:
{{"score":  75, "reasoning":  "The text is mostly fluent with natural phrasing,
though there is one minor grammatical error."}}

Your response (JSON only, no other text):"""
```

## D. Vocabulary-DFA Alignment Algorithm Details

We first formally define deterministic finite-state automata.

**Definition D.1** (Deterministic Finite-State Automaton). A deterministic finite-state automaton (DFA) is defined as a 5-tuple $A = (\Sigma, Q, q_0, \delta, F)$, where $\Sigma$ is a finite set of inputs called the alphabet, $Q$ is a finite set of states, $q_0 \in Q$ is the initial state, $\delta$ is a transition function $\delta : Q \times \Sigma \to Q$, and $F \subseteq Q$ is a set of final states.

The extended transition function $\hat{\delta}$ of $A$ is defined as the function $\hat{\delta} : Q \times \Sigma^* \to Q$ such that $\hat{\delta}(q, \epsilon) = q$ for $q \in Q$ and the empty string $\epsilon$, and $\hat{\delta}(q, wc) = \delta(\hat{\delta}(q, w), c)$ where $w \in \Sigma^*$ and $c \in \Sigma$.

We say that $a$ *accepts* a string $x \in \Sigma^*$ iff $\hat{\delta}(q_0, x) \in F$.

In Algorithm 2, CHARIFY is a function that takes as input a token $v$ and splits it into characters, which is the smallest unit of strings that is common to both the automaton and the tokenizer. For example, a tokenizer might contain the word `cat`; this token can be CHARIFYed into `c, a, t`. The idea is that the character-level automaton $A_C$ will now be able to accept this character-level string, because all transitions inside the automaton have been factorized into character-level transitions.

**Algorithm 2** Vocabulary-DFA Alignment Align$(A, V)$

**Input:** DFA $A = (\Sigma, Q, q_0, \delta, F)$, vocabulary $V$

1: // Split $A$ into char-level DFA $A_C$
2: $(C, Q_C, q_{C_0}, \delta_C, F_C) \leftarrow$ CHARSPLIT$(A)$
3: $\delta_V \leftarrow \emptyset$
4: **for** $tok \in V$ **do**
5:     // Split $tok$ into char-level sequence $c_{tok}$
6:     $c_{tok} \leftarrow$ CHARIFY$(v)$
7:     **for** $q_C \in Q_C$ **do**
8:         // Traverse $c_{tok}$ in DFA starting from $q_C$
9:         $q'_C \leftarrow$ TRAVERSE$(c_{tok}, q_C, \delta_C)$
10:        **if** $q'_C \neq$ null **then**
11:            $\delta_V \leftarrow \delta_V \cup \{(q_C, tok, q'_C)\}$
12:            $C \leftarrow C \cup tok$
13:        **end if**
14:     **end for**
15: **end for**
16: // Remove char-level transitions not in $V$
17: $\delta_V \leftarrow$ REMOVE$(\delta_V, V)$
18: **return** DFA $A_V = (C, Q_C, q_{C_0}, \delta_V, F_C)$

---

TRAVERSE in Algorithm 2 performs this operation, checking for every state in the automaton there exists a path to another state that consumes the characters in $c_{tok}$. If there is, then there is a valid path inside the automaton for the token $tok$ and we add the path as a transition.

Theorem E.1 in Section E states the correctness of Algorithm 2.

## E. Proofs

We provide simple proofs for Theorem E.1 and Theorem 3.1.

**Theorem E.1** (Vocabulary Alignment). *Let $\mathcal{L}$ be a regular constraint and $A$ be the DFA representation of $\mathcal{L}$. Then* Align$(A, V)$ *as defined by Algorithm 2 returns a DFA $A_V$ such that $A_V$ accepts a sequence of tokens $tok_1, \cdots, tok_l$ iff $tok_i \in V$ for all $1 \leq i \leq l$ and the concatenation* concat$(tok_1, \cdots, tok_l) \in \mathcal{L}$.

*Proof.* That $A_V$ accepts $tok_1, \cdots, tok_l$ if the conditions hold follows by construction of $A_V$, as the character-level automaton $A_C$ accepts CHARIFY(concat$(tok_1, \cdots, tok_l)$).

That $A_V$ accepts $tok_1, \cdots, tok_l$ *only if* the condition holds follows from the fact that (*i*) if there exists $tok_i$ such that $tok_i \notin V$, then there cannot be a transition in $A_V$ that consumes $tok_i$ by construction; and (*ii*) if concat$(tok_1, \cdots, tok_l) \notin \mathcal{L}$, the CHARIFY of this string is not accepted by $A_C$ and thus cannot be accepted by $A_V$ as well. $\square$

*Theorem 3.1* (Expected Probability). Let $A$ be a DFA that represents the regular language $\mathcal{L}$. Then Algorithm 1 returns the expected probability $\mathbb{E}_{x \sim \theta}[x \in \mathcal{L}]$.

*Proof.* By induction on the correctness of the intermediate state distributions $\mathbf{p}_i$. As the base case, $\mathbf{p}_0$ clearly represents the correct distribution as the automaton always starts from the initial state. Assume that $\mathbf{p}_i$ has the correct distribution. Then $\mathbf{p}_{i+1}$ also has the correct distribution, if the transition matrix $M_{i+1}$ is correct. By Theorem E.1, we have that $A$ accepts all possible tokenizations of a string $x \in \mathcal{L}$, which are summed in line 6 of Algorithm 1, and thus $M_{i+1}$ correctly captures the probability that $A$ can make a transition between states when accepting any tokenization of a string $x \in \mathcal{L}$. It follows from induction that $\mathbf{p}_i$ contains the correct distribution of states for every $i$, and thus that Algorithm 1 returns the expected probability. $\square$

# F. Classifier Training Specifics

We train three different classifiers. The **Full** classifier is a direct-input architecture classifier containing 10.22M parameters, and takes as input (*i*) the current PLAID latent $z_t$, (*ii*) the transformer-decoder hidden state $u_t$, (*iii*) the unigram distribution $\text{Dec}(z_t)$ (i.e., the per-position token log-probabilities), (*iv*) the target regex $\mathcal{L}$, and (*v*) the timestep $t$. The **Latent only** classifier is a direct-input architecture classifier containing 2.12M parameters, and takes as input (*i*) the current PLAID latent $z_t$, (*ii*) the target regex $\mathcal{L}$, and (*iii*) the timestep $t$. It does not take as input the transformer-decoder hidden state $u_t$ or the unigram distribution $\text{Dec}(z_t)$. The **With backbone** classifier contains 4.44M trainable classifier parameters, excluding the frozen PLAID backbone, and takes as input (*i*) the transformer-decoder hidden state $u_t$ computed by running the frozen PLAID denoiser on the current latent $z_t$, (*ii*) the target regex $\mathcal{L}$, and (*iii*) the timestep $t$. The with-backbone classifier does not take the raw latent $z_t$ or the unigram distribution $\text{Dec}(z_t)$ as direct classifier inputs; however, during guidance the gradient is backpropagated through the frozen PLAID backbone to update $z_t$. We intentionally explore different classifier configurations to make sure classifier performance is not limited due to the types of information supplied to the classifier.

We trained the classifiers only on the regexes within our benchmark. For each regex, we created 512 positive examples of length 64 tokens, matching the sequence length in our experiments. For negative samples, we used a mixture of positive samples from other benchmark regexes that do not match the current target regex, together with near-miss mutations of positive samples for the current regex. Training was performed by sampling from the positive and negative regex pools with equal probability, then noising the regexes using PLAID's own noising mechanism, following standard diffusion classifier-guidance training. The classifier saw roughly 240000 training pairs total. We note that the training setup is favourable to the classifier: it is trained precisely on the set of benchmark regexes, on examples that match exactly the sequence length used at test time, rather than a general set of regexes and sequence lengths. The trained classifiers plateaued respectively at an area-under-the-curve of 0.81 (Full), 0.74 (Latent only), and 0.79 (With backbone); we took the best-scoring checkpoint for our experiments.

# G. Detailed Benchmark Description

**JSON schema benchmarks**  For JSONSchemaBench, we begin by sampling 10 schemas from each of the 10 benchmark subcategories (`GitHub-trivial`, `GitHub-easy`, etc.), for a total of 100 schemas. We then convert each schema into a regular expression when possible. We exclude 10 schemas whose constraints are not expressible using regular languages (e.g., nested objects or unbounded lists), yielding 90 regexes.

Our current implementation of tokenizer-aligned automata (Section 3.1) is not yet optimized and can produce prohibitively large automata for certain patterns, particularly those with many wildcard transitions. To ensure tractable evaluation, we further exclude 20 regexes whose aligned automata exceed 230 states or 7.5M transitions. This results in a final set of 70 JSONSchemaBench-derived regexes.

Developing a more compact representation for tokenizer-aligned automata (e.g., using symbolic automata (D'Antoni & Veanes, 2017)) is an interesting direction for future work.

**Natural language benchmarks**  The 110 natural-language regular expressions are generated synthetically using a fixed vocabulary and a small set of template patterns. We first take the 100 most frequent English words according to (Tatman, 2018). To generate a benchmark regex in each of the following templates, we uniformly sample one or two words uniformly from this list (denoted WORD, or WORD1/WORD2 when two words are required) and a value of $n$ that satisfies the constraint:

Prefix. WORD appears exactly at the $n$-th position from the beginning of the sentence, for $1 \leq n \leq 5$.

Suffix. WORD appears exactly at the $n$-th position from the end of the sentence, for $1 \leq n \leq 3$.

Appearance. Both WORD1 and WORD2 appear somewhere in the sentence, in any order.

Between-$n$. WORD1 appears before WORD2, with exactly $n$ words in between, for $1 \leq n \leq 3$.

Between (unbounded). WORD1 appears before WORD2, with an arbitrary number of intervening words.

Word-length. The sentence contains at least one word of exactly $n$ characters, for $1 \leq n \leq 10$.

From the first five template categories, we sample 20 regexes per category. We additionally include the 10 word-length regexes (one for each length $n \in \{1, \ldots, 10\}$), resulting in 110 natural-language benchmarks in total.

For evaluation, we generate 20 samples for each natural-language benchmark and 10 samples for each JSON schema benchmark, mainly because some JSON benchmarks result in very large regexes and aligned automata that consume significant amounts of compute when generating many samples.

## H. Additional Results

*Table 3.* Constraint satisfaction and Pass@10 rates for DIFFINITY on JSON benchmarks using different padding schemes. `<json>` indicates the original JSON schema while `<EOS>` denotes the end-of-sequence token. Rates are reported with guidance scale $\gamma = 2.5$.

| | `.*<json>.*` | | `<json><EOS>.*` | | `<json><EOS>+` | |
|---|---|---|---|---|---|---|
| **Metric** | Sat. | Pass@10 | Sat. | Pass@10 | Sat. | Pass@10 |
| JSON | **68.4** | **91.4** | 56.8 | 86.5 | 33.6 | 68.8 |

*Table 4.* Comparison of results across benchmarks for GPT2-Small and Medium against DIFFINITY. Results for DIFFINITY are reported with guidance scale 2.5. Perplexity and fluency are omitted for JSON benchmarks.

| | **GPT2-Small-GCD** | | | | **GPT2-Medium-GCD** | | | | **DIFFINITY** | | | |
|---|---|---|---|---|---|---|---|---|---|---|---|---|
| **Benchmark** | Sat. | P@10 | PPL↓ | Flu.↑ | Sat. | P@10 | PPL↓ | Flu.↑ | Sat. | P@10 | PPL↓ | Flu.↑ |
| JSON | **79.3** | **98.0** | – | – | 75.3 | 97.1 | – | – | 68.4 | 91.4 | – | – |
| Prefix | 42.8 | 95.0 | 505.1 | 22.9 | 42.8 | 95.0 | 539.9 | 25.3 | **95.7** | **100.0** | **66.5** | **32.2** |
| Suffix | 19.5 | 60.0 | 123.8 | 21.5 | 17.0 | 70.0 | 115.4 | 27.2 | **96.8** | **100.0** | **59.3** | **36.8** |
| Appearance | 1.8 | 5.0 | 124.3 | 20.7 | 2.0 | 5.0 | 133.8 | 33.8 | **92.5** | **100.0** | **58.5** | **34.7** |
| Between-$n$ | 0.0 | 0.0 | – | – | 0.2 | 0.0 | 1946.8 | 5.0 | **85.5** | **95.0** | **58.7** | **33.3** |
| Between (ubd.) | 1.5 | 0.0 | 123.3 | 22.2 | 0.2 | 5.0 | 132.6 | 33.3 | **93.8** | **100.0** | **57.4** | **33.3** |
| Word Length | 88.0 | **100.0** | 312.9 | 28.8 | 90.5 | **100.0** | 169.4 | 40.2 | **95.0** | **100.0** | **60.7** | **43.3** |

*Table 5.* Comparison of average perplexity and fluency between matching, non-matching, and all samples generated by DIFFINITY at guidance scale 2.5.

| **Benchmark** | Sat rate (%) | PPL↓ | | | Flu.↑ | | |
|---|---|---|---|---|---|---|---|
| | | Match | Non-Match | All | Match | Non-Match | All |
| JSON | 68.4 | 23.4 | 28.4 | 24.8 | – | – | – |
| Prefix | 95.7 | 66.5 | 90.8 | 67.4 | 32.2 | 38.9 | 32.5 |
| Suffix | 96.8 | 59.3 | 55.8 | 59.2 | 36.8 | 36.2 | 36.8 |
| Appearance | 92.5 | 58.5 | 45.2 | 57.4 | 34.7 | 28.1 | 34.2 |
| Between-$n$ | 85.5 | 58.7 | 64.2 | 59.2 | 33.3 | 40.8 | 34.4 |
| Between (ubd.) | 93.8 | 57.4 | 50.4 | 57.0 | 33.3 | 30.0 | 33.1 |
| Word Length | 95.0 | 60.7 | 19.6 | 57.3 | 43.3 | 1.0 | 41.2 |

Table 3 compares satisfaction and pass@10 rates achieved by DIFFINITY using different schemes of padding. `.*<json>/*` allows arbitrary padding, is closest to PLAID's original training semantics, and what we used in the main text; `<json><EOS>.*` is equivalent to truncating at the first `<EOS>` token and matching the prefix against the JSON schema, most similar to the standard semantics of `<EOS>`; `<json><EOS>+` is a stricter version of the previous configuration that forces the model to emit `<EOS>` until the end of the sequence. Satisfaction rates and pass@10 decrease as the regex enforces tighter `<EOS>` semantics; we understand this as because the base PLAID model rarely emits `<EOS>` tokens during unconstrained generation (only 5.1% of samples contain at least one `<EOS>`), showing that the `<EOS>` token is quite out of distribution for the original model. Nevertheless, all three configurations show a similar trend in that DIFFINITY succeeds in constraining the base PLAID model towards outputs that satisfy complex regular constraints.

Table 4 compares results across different benchmark categories for DIFFINITY and GPT2-Small (with constrained decoding via Guidance (Guidance Contributors, 2023)). Results for GPT2-Small and Medium are largely similar to those for GPT2-Large, with GPT2-Small+GCD achieving high but comparable satisfaction rates for the JSON benchmarks, and constrained decoding affecting the natural language benchmarks particularly severely.

Table 5 provides an comparison of quality metrics (perplexity and fluency) over the samples generated by DIFFINITY, classified into those matching the regex, those that do not, and an overall aggregate. There was no clear trend showing that matching samples achieve higher quality compared to non-matching ones or vice versa, or that the matching and non-matching samples show a significant difference in terms of quality. The one outlier is word length, where non-matching samples show significantly lower perplexity *and* fluency, due to the fact that the non-matching sample set is very small (only 10 samples) that consist of degenerate samples that repeat phrases. Table 5 serves as further evidence that DIFFINITY is preserving the quality of the underlying model as constrained generation proceeds.

## I. Additional experiment showing DIFFINITY preserves the underlying probability distribution

As an additional experiment that DIFFINITY preserves the underlying distribution of the diffusion model, we test DIFFINITY and GPT2-Small-GCD on the following three regexes:

- `(The [A-Za-z .,;:"?!()-]+.|It [A-Za-z .,;:"?!()-]+.)`

- `(The first [A-Za-z .,;:"?!()-]+.|It [A-Za-z .,;:"?!()-]+.)`

- `(The first rule of Fight Club is [A-Za-z .,;:"?!()-]+.|It [A-Za-z .,;:"?!()-]+.)`

Here, the LLM is forced to select between a sentence that starts with `The` and a sentence that starts with `It`, with sentences that start with `The` getting increasingly longer prefixes. The idea is that the probability distribution between sentences that start with `The` and those that starts with `It` should be roughly equal to the following ratios:

- Probability of a sentence starting with `The` : Probability of a sentence starting with `It`

- Probability of a sentence starting with `The first` : Probability of a sentence starting with `It`

- Probability of a sentence starting with `The first rule of fight club is` : Probability of a sentence starting with `It`

Following Llama3.1-8B, the ratios of these probabilities are roughly $10.10$, $0.11$, and $1.26 * 10^{-6}$. DIFFINITY, when generating 200 samples from each regex, generates a roughly matching distribution of sentences that start with `The` and `It`: 169:30 (ratio 5.63) for the first regex, 5:176 (ratio 0.03) for the second, and 0:180 (ratio 0) for the third (for the matching regexes). The ratio of sentences starting with `The` *decreases* as the prefix grows larger, following the distribution. On the other hand, GPT2-Small-GCD fails to generate a distribution that adapts to the relative likelihood of the sentence prefixes: it generates sentences with a 'Starting with `The`':'Starting with `It`' ratio of 110:90 (ratio 1.2) for the first regex, 126:74 (ratio 1.7) for the second, and 112:88 (ratio 1.27) for the third. This illustrates how autoregressive constrained decoding *distorts* the underlying distribution: the ratios remain largely similar, dominated by the probability of the first token in the sequence.

