# OpenReview forum: "Continuous Diffusion Models Can Obey Formal Syntax"
_ICML.cc/2026/Conference — ICML 2026 regular_

### Official Review · Reviewer_xGF7 · 2026-03-10

**Soundness:** 3
**Presentation:** 3
**Significance:** 3
**Originality:** 3
**Overall Recommendation:** 4
**Confidence:** 4

**Summary:**

Continuous diffusion language models like PLAID denoise an entire sequence jointly in a continuous latent space, which makes enforcing discrete syntactic constraints during generation fundamentally harder than in autoregressive models — there is no natural left-to-right prefix to filter.

To address this, this paper proposes DIFFINITY, a training-free guidance method  by exploiting the fact that PLAID's decoder interprets any latent point as a set of independent per-position unigram distributions. Given a target regular constraint expressed as a DFA, the method constructs weighted transition matrices from the current decoder distribution and uses a dynamic program to compute the exact probability that a sampled string satisfies the constraint. The gradient of this log-probability is used as additional term to the DDPM denoising update, guiding the generation  without training any helper classifier.

A key technical contribution is a tokenizer alignment algorithm that augments the DFA to accept all valid tokenizations of strings in the language, not just canonical ones. DIFFINITY is evaluated on 180 regex constraints spanning a synthetic natural-language benchmark and a structured JSON benchmark derived from JSONSchemaBench, demonstrating strong constraint satisfaction on the former and partial coverage on the latter without any fine-tuning of the base model.

**Compliance With Llm Reviewing Policy:**

Affirmed.

**Final Justification:**

I appreciate author's effort to adequately address the issues I raised. I raised the score from weak reject to weak accept.

**Key Questions For Authors:**

1> Can you report a strict JSON evaluation with explicit EOS/PAD handling where the same validity semantics are applied to DIFFINITY and the GPT baselines — no wildcard padding, full output checked against the raw schema regex? In addition, can you report unconditional quality metrics or at least quality-satisfaction tradeoff curves? The JSON comparison is not apples-to-apples and perplexity and fluency are computed only on satisfying samples, making both the structured-generation and quality results hard to interpret. These two issues are related and fixing them together would be key to change the assessment of the paper's central claim.

2> How accurate is the approximation in Eq. (4) as a function of timestep? A late-guidance-only ablation, or a direct measurement of how well the proxy at x_t predicts final binary validity, would determine whether the conditioning story would be great theoretical grounding or post-hoc justification.

3> How essential is the factorized decoder assumption? This determines whether this is a general framework for continuous diffusion language models or a PLAID-specific technique — a distinction that directly affects how the contribution should be scoped.

**Limitations:**

I see the following limitations that the authors should consider addressing in the future:
1> the factorized decoder requirement restricts applicability to PLAID-style architectures;
2> the method is confined to regular constraints with no path to context-free ones;
3> approx. a third of JSON schemas were excluded because the current implementation cannot handle the resulting automata;
4> all quality metrics are conditional on constraint satisfaction rather than computed over all outputs.

**Strengths And Weaknesses:**

Strength:
1. The core computation is exact, not approximate: for PLAID's factorized decoder, Algorithm 1 correctly computes the acceptance probability under the decoder-induced product distribution, and the tokenizer alignment approach handles multi-character tokens in a non-trivial way  necessary to ensure correctness.

2. The method is training-free and classifier-free, making it immediately applicable to any new regex without retraining — a genuine practical advantage over typical/standard classifier guidance.

3. The natural-language benchmark results are convincingly strong: 92.9% mean satisfaction versus 20.5% for GPT2-Large-GCD, with consistently better perplexity across five of six categories and a principled explanation for why autoregressive baselines fail.

---

> ### Author Rebuttal · Authors · 2026-03-30
>
> Thank you for the review and your appreciation of our work. Additional experiments were conducted under the same conditions as in the paper unless otherwise noted.
>
> > Can you report a strict JSON evaluation with explicit EOS/PAD handling?
>
> Unfortunately, this comparison cannot be made directly because PLAID is trained to generate a fixed `seq_len` amount of tokens, using EOS as a delimiter rather than a stop token [3].
>
> We therefore report the two closest settings:
> * `<json><EOS>.*`: Equivalent to truncating at the first EOS and matching the prefix against the JSON regex
> * `<json><EOS>+`: Forces EOS padding till the end
>
> |Benchmark|Sat|Pass@10|
> |---|---:|---:|
> |`<json><EOS>.*`|56.8%|86.5%|
> |`<json><EOS>+`|33.6%|68.8%|
>
> The relatively lower satisfaction rates compared to the paper are expected, as PLAID rarely emits EOS unconditionally (5.1% of samples). Given that PLAID almost never generates JSON unconditionally (0/1000 samples in our experiments), these results show that Diffinity is effective at guiding PLAID to generate valid JSON.
>
> > Can you report unconditional quality metrics or at least quality-satisfaction tradeoff curves?
>
> We provide a comparison of geomean perplexity over matching, unmatching, and all samples:
>
> JSON benchmarks:
> |Scale|Sat|PPL Match|PPL Non-Match|PPL All|
> |---|---:|---:|---:|---:|
> |1.0|30.4%|47.3|44.8|45.6|
> |1.5|50.6%|29.4|31.3|30.3|
> |2.0|62.7%|22.5|27.7|24.3|
> |2.5|68.4%|23.4|28.4|24.8|
>
> Natural language benchmarks:
> |Scale|Sat|PPL Match|PPL Non-match|PPL All|
> |---|---:|---:|---:|---:|
> |1.0|77.3%|72.9|77.2|74.1|
> |1.5|89.1%|69.5|66.6|69.2|
> |2.0|90.8%|66.2|62.7|65.9|
> |2.5|93.0%|60.1|53.5|59.7|
>
> We also provide detailed per-category perplexity and fluency metrics for samples generated at scale 2.5:
> |Category|Sat|PPL Match|PPL Non-Match|PPL All|Fluency Match|Fluency Non-Match|Fluency All|
> |---|---:|---:|---:|---:|---:|---:|---:|
> |Prefix|95.7%|66.5|90.8|67.4|32.2|38.9|32.5|
> |Suffix|96.8%|59.3|55.8|59.2|36.8|36.2|36.8|
> |Appearance|92.5%|58.5|45.2|57.4|34.7|28.1|34.2|
> |Between-n|85.5%|58.7|64.2|59.2|33.3|40.8|34.4|
> |Between (ubd.)| 93.8%|57.4|50.4|57.0|33.3|30.0|33.1|
> |Wordlength|95.0%|60.7|19.6|57.3|43.3|1.0|41.2|
> |JSON|68.4%|23.4|28.4|24.8||||
>
> (Fluency is omitted for JSON).
>
> Overall perplexity stays stable or decreases as satisfaction improves across scales, and fluency shows no systematic drop for matching samples. This suggests Diffinity improves validity without degrading model quality.
>
> > How accurate is the approximation in Eq. (4) as a function of timestep? …
>
> We will improve the wording around Eq. (4): the decoder output (line 256) is the model’s prediction of the clean text $\hat{x}_0$ given the noisy latent $x_t$, and thus the question is how well the model can predict $x_0$ given $x_t$, not how well the noisy $x_t$ directly approximates $x_0$.
>
> We did conduct an ablation study where the guidance was only applied at the final 25% or 50% of timesteps. We used the JSON benchmarks, using `<json><EOS>.*` (from the strict JSON evaluation) as the regex pattern.
>
> |Guidance window|Sat|Pass@10|
> |---|---:|---:|
> |Final 25%|21.0%|64.4%|
> |Final 50%|36.2%|75.0%|
> |Full|56.8%|86.5%|
>
> The ablation shows that Diffinity provides a meaningful signal even at early timesteps and is not just a late-stage heuristic.
>
> > How essential is the factorized decoder assumption? This determines whether this is a general framework for continuous diffusion language models or a PLAID-specific technique
>
> The ‘factorized decoder’ assumption holds true for most modern text diffusion models [1-4], not just for PLAID. Precisely speaking, our assumption is not that the decoder is factorized, but that the model provides its prediction of the clean output ($\hat{x_0}$) given the current noisy latent $x_t$ as a unigram distribution at each timestep. This unigram distribution is simple, and also mathematically sufficient in the diffusion-process sense: the true sequence distribution $p(x_0)$ is induced by composing the per-position token marginals over the timesteps, rather than requiring the model’s guess $\hat{x_0}$ to represent a joint distribution directly at each timestep. Thus, our assumption is a principled parametrization common to most modern implementations of text diffusion models [1-4], rather than a PLAID-specific assumption.
>
> > the method is confined to regular constraints with no path to context-free ones
>
> Because diffusion models require a fixed `seq_len` for generation, it is possible to constrain the length of strings in a CFG, resulting in a regular language. Previous work on constrained decoding for discrete diffusion [6] has investigated efficient ways to impose this length constraint on CFGs; a potential path forward would be to extend these ideas to efficiently compute expected probabilities.
>
> ### References
> [1] arxiv.org/abs/2502.09992
>
> [2] arxiv.org/abs/2508.15487
>
> [3] https://openreview.net/forum?id=VGv5y60sXC
>
> [4] https://arxiv.org/abs/2305.18619
>
> [5] https://openreview.net/forum?id=7Sph4KyeYO

---

> > ### Author Rebuttal · Reviewer_xGF7 · 2026-04-02
> >
> > I appreciate author's effort to adequately address the issues I raised. I raised the score from weak reject to weak accept

---

### Official Review · Reviewer_1ypT · 2026-03-12

**Soundness:** 3
**Presentation:** 2
**Significance:** 2
**Originality:** 3
**Overall Recommendation:** 4
**Confidence:** 1

**Summary:**

This paper studies how to enforce formal syntactic constraints for continuous diffusion language models, where standard constrained decoding is not directly applicable because generation proceeds in a continuous latent space rather than through an explicit discrete prefix. The paper proposes DIFFINITY, a training-free guidance method for steering continuous diffusion models toward outputs that satisfy regex-defined constraints.

**Compliance With Llm Reviewing Policy:**

Affirmed.

**Final Justification:**

I thank the authors for their detailed response. I believe my current score remains appropriate.

**Key Questions For Authors:**

1. The paper shows very strong gains on synthetic natural-language regexes but weaker gains on JSON. What do the authors believe is the main bottleneck on JSON: base-model capacity, automaton size, sequence length, or the guidance approximation itself?
2. Since 20 JSON regexes were excluded due to large aligned automata, how much does current performance depend on this filtering? Do the authors expect the method to remain practical on realistic large-schema workloads without symbolic or compressed automata?

**Limitations:**

yes

**Strengths And Weaknesses:**

### Strengths
1. The paper addresses a real and well-motivated gap: formal-syntax control for continuous diffusion LMs is much less developed than constrained decoding for autoregressive models or masking-based discrete diffusion models.
2. The tokenizer–automaton alignment component is a genuine contribution rather than a minor implementation detail
3. The paper includes useful ablations and comparisons beyond the main table

### Weaknesses
To be honest, I am not familiar with the core idea involved in the paper. Perhaps for professionals in the field, this is an excellent paper, so I am willing to refer to other reviewers' opinions to adjust my score. The following are just some of my personal suggestions.

1. The JSON results are not especially strong relative to the autoregressive baselines. DIFFINITY is clearly interesting, but on JSON schema benchmarks it is still below GPT2-GCD in both satisfaction rate and pass@10.
2. Maybe the computational cost is a serious limitation. The method requires differentiating through automaton-based dynamic programming, and the latency scales roughly linearly with the number of transitions; batching helps only modestly.

---

> ### Author Rebuttal · Authors · 2026-03-30
>
> Thank you for the review and your appreciation of our work.
>
> > The paper shows very strong gains on synthetic natural-language regexes but weaker gains on JSON. What do the authors believe is the main bottleneck on JSON: base-model capacity, automaton size, sequence length, or the guidance approximation itself?
>
> The answer is, as the reviewer guesses, the base model’s capacity: PLAID was trained to generate coherent natural language sequences and does not generate any valid JSON (0/1000 samples in our experiments) under unconstrained generation. Diffinity applies a gradient to guide the model towards generating JSON, but if the base model’s own updates are far from generating valid JSON, the model can still generate non-JSON samples.
>
> > Since 20 JSON regexes were excluded due to large aligned automata, how much does current performance depend on this filtering? Do the authors expect the method to remain practical on realistic large-schema workloads without symbolic or compressed automata?
>
> [SCLB]
>
> The cutoff criterion from the paper is primarily a memory limitation: in which we can fit 20 samples on our GPU (48GB). The main weakness of Diffinity currently is memory consumption. The expected probability computation is currently implemented as matrix multiplications over two major matrices: the transition matrix of the token-aligned DFA, and the reduction matrix which translates the unigram distributions from the model into the transition matrix. The transition matrix consumes memory on the order of `|States|^2`, while the reduction matrix consumes memory on the order of `|Transitions|`, which becomes problematic for very large automata.
>
> On the other hand, we performed (after the submission) some basic optimizations on Diffinity to take better advantage of CUDA operations during the review period, and latency is substantially improved compared to the numbers reported in the paper. Under the same experimental settings (64 sequence length, 1024 timesteps, 20 samples), per sample latency is now between 11.9s (for our smallest regex, with 71000 transitions) and 45s (for the largest regex that will allow 20 samples in memory, with 2.5M transitions), which represents up to a 5 times speedup compared to unoptimized Diffinity. The larger automata are from the harder natural language benchmarks (between-n with a high n) and medium-difficulty JSON benchmarks. Unconditional generation in PLAID has a per-sample latency of around 3s under the same conditions.
>
> We do note that a ‘complex regex’ in the ordinary sense does not necessarily translate to a complex automaton in our setting: our statistics are dominated mostly by transition count, for which wildcard matches are a very significant contributor due to the tokenizer alignment step. Thus, relatively simpler regexes such as `.* for .* .* .* and .*` (as in our between-n benchmarks) can result in large automata due to the high number of wildcards, while very complex regexes that severely limit the set of satisfying strings can actually result in automata that are much smaller. Thus the question of whether Diffinity can be immediately used in realistic workloads depends quite a bit on the type of the workload.
>
> To summarize, Diffinity is currently practically usable, though not extremely efficient, for moderate workloads; workloads involving very large automata would primarily require optimizations to the in-memory representation of the computations to become practical. Compressed or symbolic automata are one promising solution to minimize memory footprint; there may be other ways (such as denser tensor representations in memory) as well.

---

> > ### Author Rebuttal · Reviewer_1ypT · 2026-04-01
> >
> > I thank the authors for their detailed response. I believe my current score remains appropriate.

---

### Official Review · Reviewer_ZJL7 · 2026-03-15

**Soundness:** 2
**Presentation:** 2
**Significance:** 2
**Originality:** 2
**Overall Recommendation:** 3
**Confidence:** 4

**Summary:**

This paper introduces a training-free guidance framework that biases continuous diffusion models towards syntax constraints. The core insight lies in modeling the expected probability using a DFA, which is then injected into the sampling process, akin to classifier guidance. Experiments were performed using PLAID on tasks including JSON and natural language format satisfaction, showing that the proposed approach can enhance constraint satisfaction and exhibits tradeoffs similar to traditional guidance-based sampling approaches for diffusion models.

**Compliance With Llm Reviewing Policy:**

Affirmed.

**Final Justification:**

continuous diffusion has a unique capability absent from modern, unmasking discrete diffusion models - iterative refinement, the ability to revise decisions made early on in the denoising process

This may not be true. The same argument also holds for discrete diffusion models and there have been plenty of works developing iterative refinement techniques for discrete diffusion. e.g., [1] [2]

[1] Wang et al. Remasking discrete diffusion models with inference-time scaling. NeurIPS 2025.

[2] Ou et al. Inference-Time Scaling of Discrete Diffusion Models via Importance Weighting and Optimal Proposal Design. ICLR 2026.

Furthermore, there is still no comparison/discussion about DFA-based constrained decoding literature in AR models, and the key confusing part in the motivation, i.e., why using continuous diffusion on a naturally discrete task (language modeling), is still not clarified or justified by empirical comparisons. Therefore, I maintain the score.


---------------Update--------------

Score raised to 3 after another round of discussion with the authors.

Still, I encourage the readers to pursue further in this direction and add more concrete and systematic comparisons. This, from my perspective, would be really important in amplifying the scope and impact of this work in the community.

**Key Questions For Authors:**

1. Why does the author target on continuous language models? Investigating constrained decoding/constraint satisfaction problem for discrete diffusion language models would be more reasonable.

2. The experiments are only conducted on top of PLAID, which is not a very well performing language model. The scale of the experiments is also limited which raises concern on the scalability of the approach. More experiments on larger models and more state-of-the-art models would make the contribution much stronger.

3. There is no baselines compared in the paper. Methods, at least like classifier guidance, should be considered and compared.

**Limitations:**

I did not find discussions on limitations in the paper. The authors should address the limitations with respect to, e.g., relying on the continuous diffusion framework instead of discrete diffusion for language modeling.

**Strengths And Weaknesses:**

## Strengths

1. The paper tackles an interesting problem of biasing the sampling of diffusion language models towards constraint satisfaction.

2. The proposed approach is well-motivated and the method is very easy to follow. The overall presentation of the paper is clear.

## Weaknesses

1. The overall approach seems to be a direct instantiation of DFA-based constrained decoding framework for autoregressive language models in the diffusion sampling context. A more detailed description on the conceptual distinction with respect to these previous works in AR language model literature should be addressed.

2. The framework is mainly proposed under a continuous diffusion language model setting, on top of the PLAID model. However, discrete diffusion models instead exhibit strong performance these days and have become the core interest when it comes to applying diffusion models to language/sequence modeling. It is unclear whether the proposed approach can be extended for discrete diffusion language models as well, where a family of state-of-the-art discrete diffusion models on large-scale and more significant benchmarks is available.

3. There are no baselines on diffusion language models considered. Minimally, classifier-guidance should be implemented and compared.

---

> ### Author Rebuttal · Authors · 2026-03-30
>
> Thank you for the review and your assessment of our work. We have conducted additional experiments to answer concerns, and would like the chance to correct some misconceptions the reviewer may have had.
>
> > The overall approach seems to be a direct instantiation of DFA-based constrained decoding framework for autoregressive language models in the diffusion sampling context.
>
> First, we would like to emphasize that our method is very different from constrained decoding on AR models. In AR constrained decoding, the key object is a binary token mask which filters next-token choices; however, as stated in the second paragraph of the intro, continuous diffusion does not have the concept of a next token. The key object in our approach is thus the expected probability that the model will generate a sample that matches the target regex. Both our approach and AR constrained decoding operate over DFAs as a means to compute their respective key objects; not because the objects are the same.
>
> In particular, the token masks for AR constrained decoding can severely distort the conditional distribution of strings that satisfy the constraint. On the other hand, our approach is directly designed to sample from the desired conditional distribution (Section 3.3), and does so far better in practice (Appendix G.1). Preserving this conditioned distribution is important because it is precisely the object that captures the model’s own learned knowledge about which constraint-satisfying outputs are most likely; distorting this distribution causes sampling to produce outputs that are no longer aligned with the model’s beliefs.
>
> > Why does the author target on continuous language models? Investigating constrained decoding/constraint satisfaction problem for discrete diffusion language models would be more reasonable.
>
> While it is true that discrete diffusion models have recently received more attention, continuous diffusion has a unique capability absent from modern, unmasking discrete diffusion models [1, 2] - iterative refinement, the ability to revise decisions made early on in the denoising process [3]. Recent papers [3] have recognized this ability as an advantage of continuous diffusion and have shown competitive results in small-scale experiments. We believe this capability, when combined with external analyses to guide the refinement process (e.g., the expected probabilities), is a promising approach to generate high quality outputs even from models of smaller scale. We agree that extending our method to discrete diffusion is an important future direction, but do not believe that our focus on continuous diffusion invalidates our contribution.
>
> From a technical perspective, the continuous latent space makes it difficult to enforce formal syntax as the internal state is non-discrete. This problem is precisely what this paper tackles, enabling outputs adhering to formal syntax for an underexplored but promising class of language models.
>
>
> > It is unclear whether the proposed approach can be extended for discrete diffusion language models as well…,
>
> The core concept of using the expected probability as a classifier signal applies to discrete diffusion as well. The challenge is that, instead of adding a gradient, discrete diffusion must sample $x_{t-1}$ from a reverse kernel proportional to the condition r: $p(x_{t-1}\mid x_t, r) \propto \mu_\theta(x_{t-1}\mid x_t) \cdot s(x_{t-1})$. Our expected probabilities can replace the classifier score $s(x_{t-1})$, but it is an open problem how to compute this reweighted distribution efficiently.
>
> > The experiments are only conducted on top of PLAID, which is not a very well performing language model. The scale of the experiments is also limited which raises concern on the scalability of the approach.
>
> We would like to answer this question in two steps: on why the evaluation on PLAID is still valid, and the concern about scalability.
> * PLAID as the evaluation model for Diffinity
>
> We agree that an evaluation on larger and newer models would strengthen the paper. However, our goal is to show that formal regex constraints can be imposed effectively on continuous diffusion language models, rather than to claim state-of-the-art generation quality, and PLAID is a suitable model for demonstrating this goal. Our evaluation shows that Diffinity significantly improves constraint satisfaction while preserving the base model’s quality, as reflected in the small perplexity and fluency gaps, supporting the our core contribution: a flexible guidance mechanism for continuous diffusion LMs that can be applied to future models as they become available.
> * Scalability
>
> Please see section [SCLB] for Reviewer 1ypT for a discussion on scalability.
> > Methods, at least like classifier guidance, should be considered and compared
>
> Please see Review jbAZ for a discussion on a classifier baseline.
> ### References
> [1] arxiv.org/abs/2502.09992
>
> [2] arxiv.org/abs/2508.15487
>
> [3] https://openreview.net/forum?id=VGv5y60sXC

---

> > ### Author Rebuttal · Reviewer_ZJL7 · 2026-04-03
> >
> > I thank the authors for the rebuttal. However, the key concerns still remain.
> >
> > **continuous diffusion has a unique capability absent from modern, unmasking discrete diffusion models - iterative refinement, the ability to revise decisions made early on in the denoising process**
> >
> > This may not be true. The same argument also holds for discrete diffusion models and there have been **plenty of** works developing iterative refinement techniques for discrete diffusion. e.g., [1] [2]
> >
> > [1] Wang et al. Remasking discrete diffusion models with inference-time scaling. NeurIPS 2025.
> >
> > [2] Ou et al. Inference-Time Scaling of Discrete Diffusion Models via Importance Weighting and Optimal Proposal Design. ICLR 2026.
> >
> > Furthermore, there is still no comparison/discussion about DFA-based constrained decoding literature in AR models, and the key confusing part in the motivation, i.e., why using continuous diffusion on a naturally discrete task (language modeling), is still not clarified or justified by empirical comparisons. Therefore, I maintain the score.

---

> > > ### Author Response · Authors · 2026-04-08
> > >
> > > Thanks for the response. We seem to have misunderstood some of the reviewer’s original concerns. We can answer the reviewer’s questions both qualitatively and quantitatively.
> > >
> > > # Diffinity vs. Discrete diffusion
> > >
> > > Our claim is that continuous diffusion+Diffinity can be more effective in generating quality outputs satisfying formal syntax than constrained decoding on discrete diffusion. As evidence, we compare Diffinity against LLaDA-8B-Base+DINGO [3,4], a state-of-the-art baseline for constrained decoding on discrete diffusion. We used the prompts `Generate english text: ` for the natural language benchmarks and `Generate json: ` for the JSON benchmarks (LLaDA is promptable and empty prompts yield degraded outputs). Unconstrained LLaDA has lower geomean perplexity (23.9 for the prompt `Generate english text: `) compared to PLAID (61.6). We used the default settings for DINGO provided in their artifact [4] to apply constrained decoding over LLaDA on our benchmark regexes. The table reports satisfaction rate, geomean perplexity, and fluency on satisfying samples.
> > >
> > > |Benchmark|Diffinity Sat.|PPL|Flu.||LLaDA+DINGO Sat.|PPL|Flu.|
> > > |---|---:|---:|---:|---|---:|---:|---:|
> > > |Prefix|95.7|66.5|32.2||100.0|170.1|3.8|
> > > |Suffix|96.8|59.3|36.8||100.0|219.2|3.4|
> > > |Appearance|92.5|58.5|34.7||99.8|252.5|3.8|
> > > |Between-n|85.5|58.7|33.3||100.0|253.7|3.7|
> > > |Between (ubd.)|93.8|57.4|33.3||95.5|234.5|3.6|
> > > |WordLength|95.0|60.7|43.3||100.0|217.7|4.0|
> > > |Natural (all)|93.0|60.1|35.1||99.1|222.8|3.7|
> > > |JSON|68.4|23.4|-||63.1|63.7|-|
> > >
> > > LLaDA+DINGO attains high satisfaction rates, but with *much worse perplexity and fluency* than Diffinity. Constrained decoding markedly _degraded_ LLaDA’s output quality. These results support our claim: it is too soon to conclude that discrete diffusion is more natural or reasonable for text generation. Despite using a weaker base model, continuous diffusion+Diffinity generates higher-quality constraint-satisfying sample than LLaDA+DINGO in our experiments.
> > >
> > > > plenty of works developing iterative refinement techniques for discrete diffusion
> > >
> > > We thank the reviewer for the references. We agree that ReMDM and SMC-based sampling are promising steps toward iterative refinement in discrete diffusion. However, these methods are primarily sampler modifications: the base model remains an unmasking model, and iterative refinement is introduced by altering the reverse process (remasking or particle-based resampling). As a result, they alter the distribution the model generates at inference time, and good results on one model does not imply a general solution applicable to other models.
> > >
> > > As evidence, we implemented and compared ReMDM-style remasking on LLaDA-8B [3] to standard LLaDA on the prompt `Generate english text: `. ReMDM decreased mean perplexity from 23.9 to 20.0, but increased degenerate samples (e.g., prompt-repeating samples) by 10% of the full sample set, showing that sampler-level refinement does not generalize to immediate performance gains across different models.
> > >
> > > Moreover, it is unclear how to perform constrained decoding for such modified discrete samplers, especially while preserving the conditional distribution as Diffinity does. Diffinity instead provides a general solution applicable for all continuous diffusion models exposing per-timestep unigram distributions.
> > >
> > > # Diffinity vs. Autoregressive (AR) constrained decoding
> > >
> > > In case there was a misunderstanding, we’d like to draw the reviewer’s attention to Table 1 and 2 in Section 4, which compare Diffinity against DFA-based AR constrained decoding on GPT2-Small/Medium/Large. Diffinity is comparable to AR constrained decoding for the JSON benchmarks and significantly outperforms it on natural language benchmarks in terms of satisfaction rate / perplexity, while maintaining similar fluency to GPT2-Large. These results are remarkable, as the base PLAID model has performance in between GPT2-Small and Medium [1]. Our experiments highlight the two main drawbacks of AR constrained decoding:
> > >
> > > * AR constrained decoding distorts the target conditional distribution $p(x|r)$ given a constraint $r$ [2], whereas Diffinity converges towards $p(x|r)$ as the timesteps size shrinks (Section 3.3). This gap is illustrated by the severe perplexity gap between matching samples (Table 2), and Appendix G.1.
> > > * AR constrained decoding often cannot enforce the `max_tokens` parameter: the token mask tells us which tokens have an accepting continuation, but not _when_ we must enter an accepting state (lines 294-302). This limitation is why AR constrained decoding has very low satisfaction rates on the natural language benchmarks. Enforcing `max_tokens` is possible in theory by DFA intersection, but is computationally expensive and unsupported by many implementations. Diffinity, on the other hand, supports `max_tokens` natively.
> > >
> > > ## References
> > > [1] arxiv.org/abs/2305.18619
> > >
> > > [2] arxiv.org/abs/2405.21047
> > >
> > > [3] huggingface.co/GSAI-ML/LLaDA-8B-Base
> > >
> > > [4] openreview.net/forum?id=KaYMGsnZ4R

---

### Official Review · Reviewer_jbAZ · 2026-03-16

**Soundness:** 2
**Presentation:** 3
**Significance:** 3
**Originality:** 3
**Overall Recommendation:** 4
**Confidence:** 4

**Summary:**

This paper tackles training-free guidance for continuous-space text diffusion. The paper vectorizes the acceptance computation for regular languages. Given a language, the approach constructs a vocabulary-aligned DFA. This DFA and the unigram marginals of the text diffusion model define a trellis over token sequences, where the probability of acceptance can be approximated via a sequence of matrix-vector products. As this is fully differentiable, it can be used for classifier guidance.

Results are promising compared to explicit decoding-based guidance in an autoregressive model. While there is some gap between the method on complex JSON generation, perhaps this gap disappears with more powerful models.

**Compliance With Llm Reviewing Policy:**

Affirmed.

**Key Questions For Authors:**

The main missing experiment is comparing to a trained classifier, which may have different accuracy and computational tradeoffs than the proposed method.

Technically, the method works on any text diffusion model where unigram marginals are available, and constructs a classifier without training. This is also applicable to eg text diffusion models with uniform noise.

**Limitations:**

Yes

**Strengths And Weaknesses:**

This paper does well on presentation and originality. The paper was easy to understand, and prior work such as [Simple Guidance Mechanisms for Discrete Diffusion Models](https://arxiv.org/abs/2412.10193) relied on trained classifiers. However, for soundness and significance, I believe that a comparison to a classifier is missing. It may be that training a classifier is actually computationally cheaper than simulating the DFA on the marginals, since the vectorized DFA is computationally expensive: $T \times V \times V$.

---

> ### Author Rebuttal · Authors · 2026-03-30
>
> Thank you for the review and your appreciation of our work.
>
> Our main reason for not including a comparison with classifier guidance was because regex satisfaction is a _binary_ signal—a sample either satisfies a regex, or it does not—which we conjectured would be ill-suited for classifier guidance.
> To make the comparison explicit, we have trained a classifier and will include a comparison with the classifier as part of our revision. Classifier guidance was only able to achieve an average satisfaction rate of 1.34% and pass@10 rate of 5.91% over our benchmarks on the best hyperparameter settings, notably achieving 0% on the JSON benchmarks (which matches the behavior of the base PLAID model). Successes for natural language benchmarks were concentrated in the Word Length category, where the constraint is that the generated sentence contains at least one word of length $n$ for $1 \leq n \leq 10$: often satisfiable by the model without external guidance. A table summarizing the results is provided below:
>
>   | Category | # Regexes | Pass@10 | Avg. Satisfaction Rate |
>   |---|---:|---:|---:|
>   | Prefix | 20 | 0.00% | 0.00% |
>   | Suffix | 20 | 0.00% | 0.00% |
>   | Appearance | 20 | 0.00% | 0.00% |
>   | Between-n | 20 | 0.00% | 0.00% |
>   | Between (ubd.) | 20 | 5.00% | 0.25% |
>   | Wordlength | 10 | 100.00% | 24.50% |
>   | Natural total | 110 | 10.00% | 2.27% |
>   | JSON total | 70 | 0.00% | 0.00% |
>   | Overall | 186 | 5.91% | 1.34% |
>
> Compared with results from Diffinity, which achieves an average of 68.4% and 93.1% satisfaction rates on JSON and natural language benchmarks, these results suggest that training a classifier to guide diffusion models to satisfy formal discrete constraints, like regular expressions, is unrealistic. We provide more details about our classifier below.
>
>
> ## Classifier specifics
>
> Our classifier contains 10.22M parameters and takes as input i) the current PLAID latent `z_t`, ii) the transformer-decoder hidden state `u_t`, iii) the per-position token log-probabilities `log_probs_t`, iv) the target regex `r`, v) and the timestep `t`. We intentionally gave the classifier explicit access to the current latent, the transformer hidden state, and the token marginals, to ensure that all relevant information was presented to the classifier in a learnable form, reducing the probability that classifier performance is limited due to withholding relevant information.
>
> We trained the classifier _only_ on the regexes within our benchmark. For each regex, we created 512 positive examples of length 64 tokens, matching the sequence length in our experiments.  For negative samples, we used a mixture of positive samples from other benchmark regexes that do not match the current target regex, together with near-miss mutations of positive samples for the current regex. Training was performed by sampling from the positive and negative regex pools with equal probability, then noising the regexes using PLAID’s own noising mechanism, following standard diffusion classifier-guidance training. The classifier saw roughly 240000 training pairs total. We note that the training setup is favourable to the classifier: it is trained precisely on the set of benchmark regexes, on examples that match exactly the sequence length used at test time, rather than a general set of regexes and sequence lengths.
>
> The trained classifier plateaued at an AUC (area-under-the-curve) of 0.81, and we took the best-scoring checkpoint for our experiments. The relatively high AUC score indicates that the classifier did learn a meaningful signal: it is capable of distinguishing, given a noised example, whether the underlying clean sample satisfies the target regex. However, this classification ability did not translate into effective guidance at generation time, as demonstrated by our results. These results show that merely taking the gradient of the classifier, which learns a score based on the binary truth of whether a sample satisfies a regex or not, does not translate into a useful guidance signal for our target problem.

---

### Decision · Program_Chairs · 2026-04-30

**Decision:**

Accept (regular)

**Comment:**

The paper introduces Diffinity, a training-free guidance method for enforcing formal syntactic constraints in continuous diffusion language models. Reviewers find the approach technically interesting and well-motivated. There are consistent concerns about limited evaluation and scope, including reliance on a single relatively weak base model, lack of comparisons to key baselines (initially), and unclear positioning relative to prior constrained decoding methods. The rebuttal strengthens the paper by adding classifier and discrete diffusion comparisons and clarifying technical distinctions, but some concerns about scope and generality remain.